# Mediating Effects of Job Satisfaction and Organizational Commitment between Problems with Performance Appraisal and Organizational Citizenship Behavior

**Khahan Na-Nan [1,*], Suteeluck Kanthong [1], Jamnean Joungtrakul [2] and Ian David Smith [3]**

1   Faculty of Business Administration, Rajamangala University of Technology Thanyaburi, Thanyaburi 12110, Thailand; suteeluck_k@rmutt.ac.th
2   College of Social Sciences, Far East University, Eumseong-gun, Chungcheongbuk-do 27601, Korea; drjamnean@kdu.ac.kr
3   School of Education & Social Work, The University of Sydney, Sydney 2006, Australia; ismith12@bigpond.com
*   Correspondence: khahan_n@rmutt.ac.th

**Abstract:** The objective of this study was to test the developed models with empirical evidence regarding job satisfaction (JS) and organizational commitment (OC) as mediators and influential variables concerning problems with performance appraisal and organizational citizenship behavior (OCB). The sample consisted of 450 employees who worked at the operational level in automobile parts manufacturing companies in the Navanakorn Industrial Estate, Thailand. The study's instrument was applied from previous research and verified for content validity and reliability before collecting the data. Structural equation analysis of 5000 rounds of bootstrapping was used to examine the model developed from the empirical data and the influence of problem variables in appraising work performance. The developed model fitted well with the empirical evidence. JS and OC were identified as mediators among problem variables in appraising performance and OCB, while problem variables in appraising work performance influenced OCB through JS and OC. Findings from this study expand our knowledge of this field and are useful for effective human resource management and performance appraisal. The developed model explains the phenomena of problems with performance appraisal concerning fairness, JS, OC, and OCB. JS and OC are useful mediators to explain and expand knowledge in human resource management and performance appraisal.

**Keywords:** job satisfaction; organizational commitment; problems with performance appraisal; organizational citizenship behavior

## 1. Introduction

Organizational citizenship behavior (OCB) is valued by educators and practitioners [1] and is manifested in five dimensions as altruism, conscientiousness, sportsmanship, courtesy, and civic virtue [2]. These dimensions enhance employees' desirable behavior and work performance according to the expectation of administrators and organizations [3].

Previous research has suggested that OCB acts as a mediator for job satisfaction (JS) [4,5] and organizational commitment (OC) [6,7]. OCB has been defined as people's mental perception in assessing or perceiving different matters concerning their work responsibility. Indarti et al. [8], Devece et al. [9], and Prasetio et al. [10] agreed that employees manifest OCB if administrators build their JS and OC. This relationship takes the form of cause and effect. JS [11,12] and OC [10–13] were identified as factors

influenced by other variables and function as moderators or mediators. Therefore, to promote OCB, the two factors of JS and OC must be studied.

Performance appraisal is important for effective human resource management [14,15]. Performance appraisal is used to assess whether employees' work performance is effective, according to company expectations. In general, large organizations or government sectors use performance appraisals to make decisions on the promotion and retention of employees [16]. Moreover, educators and researchers in human resource management and development are very interested in fair performance appraisal because it is an important factor that affects employees' JS, OC, and OCB. By contrast, if the appraisal is unfair, this will reduce employees' JS [17], level of OC [18], and level of OCB [19]. Therefore, in this study, factors of OCB were studied to determine whether they were influential in particular changing contexts and situations to raise awareness for improving and developing appraisal effectiveness.

Performance appraisal is complex in practice. Although related working units attempt to find ways to build clear indicators of work performance and behavior, problems often arise between employees and organizations [20], resulting in job dissatisfaction, lack of OC, bad behavior, and reduction of work performance [21]. Researchers in human resource management and development pay a lot of attention to problems of performance appraisals. The examples of the problems include appraisers' lack of important facts, unclear standards for performance appraisals, appraisers' inattention to importance of appraisals, appraisers' unpreparedness for reviewing employee performance appraisals, appraisers' dishonest and insincere appraisals, appraisers' lack of appraising skills, employees not being informed about their appraisal results, organizations' lack of appropriate systems for rewards and penalty to support appraisals, no discussion between appraisers and appraisees, and appraisers' unclear appraisals. These factors cause appraisal problems which are mostly from the person and appraisal process. As a result, such performance appraisals are ineffective and result in employees' negative attitudes towards their organizations. If these problems continue, employees will repeat such behaviors when they are promoted to higher positions as administrators and appraisers. This will become a vicious circle which causes employees' job dissatisfaction [17], reduction of organizational engagement [19], reduction of organizational citizenship behaviors [22], and organizations' inability to compete effectively with industrial rivals, because capable employees are discouraged and eventually leave their organizations.

Because of these problems, the researchers were interested in studying direct effects of problem factors in performance appraisals on citizenship behaviors, and indirect effects through mediators of job satisfaction on factors of employees' organizational engagement.

## 2. Theories for Explaining the Concepts and Study Framework

Based on the equity theory developed by Adams [23], people often compare themselves with others. If employees are treated equally, they will have positive attitudes toward their work and manifest effective working behavior. By contrast, if they are treated unfairly, negative attitudes and behavior occur, such as a reduction of JS [24,25], lack of OC [26,27], and low OCB [28–30]. Accordingly, fairness is regarded as the main cause of positive attitudes toward work and the desirable behavior expected by administrators or organizations.

However, unfairness often occurs, especially in performance appraisal through problems with the appraisal process and problems with the appraising person [31]. These problems affect attitudes and behavior. In this study, performance appraisal affecting OCB was assessed based on the explanations of equity theory. If performance appraisal is unfair, employees' OCB will be low.

### 2.1. Organizational Citizenship Behavior

Organ [32] defined OCB as the behavior of each person, occurring from his/her consideration and free decision-making to do something by themselves. Such behavior is not directly affected by rewards, and is not clearly prescribed by companies. Organ [32] and Podsakoff et al. [33] classified five components of OCB: (1) Altruism refers to behavior caused by one's own free decision-making to help

others solve different work problems; (2) Conscientiousness refers to behavior to freely make decisions to perform work better than the minimum level required, such as attention to work, conformance to rules and regulations, use of time off, and other work behavior; (3) Sportsmanship refers to employees' willingness to be patient in various situations without expressing dissatisfaction, such as avoidance of complaints, gossip, blame, and trivialities; (4) Courtesy refers to people's behavior when making decisions to prevent relationship problems from working with others. They must consider that their own actions may impact others, and always give respect to the personal rights of others; and (5) Civic virtue refers to participating in various work-related activities.

Podsakoff et al. [34] reviewed related research and analyzed the factors influencing OCB. They determined that the factors of job satisfaction, commitment, and work attributes were predictors of OCB. Similarly, Mogotsi et al. [35] developed a model to test the relationships between the variables of teachers' satisfaction and commitment to citizenship behavior. They found that these two variables could explain 76% of good citizenship behavior, while O'driscoll et al. [36] found that the working environment influenced OCB. They studied the same variables with different sample groups and discovered that work attributes, work environment, job satisfaction, and organizational commitment were all related to and influenced OCB.

## 2.2. Problems with Performance Appraisal

Adams [23] proposed that equity occurs when management is effective, transparent, and checkable, whereas inequality occurs when management is ineffective, not transparent, and unable to be checked. According to Gilliland and Langdon [37], fairness includes three aspects: (1) Procedural justice is fairness perceived by employees about the appropriateness of appraisal procedures in which the appraisees can express their opinions and receive feedback; the appraisal results are transparent and reliable, and the judgment procedure is not biased with double standards [38]; (2) Interpersonal fairness is people's perception of the practice during the appraisal period, clear communication, honesty, ethics, and clear targets of performance appraisal [39]; and (3) Outcome fairness involves satisfaction of appraisal results that are comparable to work output. If the results are below employee expectations, a sense of unfairness will occur [40].

Na-Nan et al. [31] studied problems with performance appraisal based on the theory of justice and the problems with performance appraisal concepts. Exploratory factor analysis found that the problems with performance appraisal can be divided into two components. The first component is problems with the appraisal process, referring to a lack of clear targets for what to appraise and how to use the appraisal results, lack of reliable appraisal forms that create confusion with the appraisal system, use of abstract appraisal forms without validity and reliability, use of old-fashioned appraisal forms, appraisal indicators that are unclear and do not reflect the context of the appraised job, an inappropriate period of time (i.e., too long or too quick) to work effectively, and lack of involvement between employees and administrators. The second component is problems with the appraising person, referring to administrators not giving importance to real performance issues, appraisers' lack of knowledge and understanding of the appraisal, lack of fairness with bias in the appraisal, setting too high criteria, not giving feedback to the appraisees, appraisees' lack of knowledge and understanding of the reasons for performance appraisal, and not knowing about the targets and objectives of the appraisal. The two components were tested by confirmatory factor analysis, their study demonstrating that their empirical data were consistent with the two-component problems of performance appraisal.

The concepts and theory of justice can be used to explain performance appraisals, since the performance appraisals are based on justice of the processes designed by organizations and people who perform as appraisers, leading to dissatisfaction and negative results on employees' behaviors and organizations. Therefore, appraisal problems are related to justice in the appraisal process and the people who appraise employees' performance.

Based on performance, appraisers often focus on explaining the emerging problems using related concepts and theories. Accordingly, problems with performance appraisal can be classified into two

main types: (1) Problems with the performance appraisal process include abstract appraisal indicators relying on appraisers' judgment, lack of stakeholders' involvement, old-fashioned and ineffective appraisal forms, unfairness of appraisal, inappropriate period of appraisal time, and discontinuity of the appraisal [41]. According to Paul [42], problems with the appraisal process include inappropriateness of appraisal criteria and data sources. Grund and Przemeck [43] also studied problems with performance appraisal. They found that abstractness of the appraisal criteria or indicators may lead to appraisers' bias on each appraisal, resulting in appraisees' dissatisfaction. In some organizations, appraisal forms do not conform to the work contexts; they are so complicated that the appraisers are confused and do not understand the appraising methods, and the appraisal does not reflect the actual performance of the employees [44]; (2) Problems with the performance appraisal result from the appraisers' lack of knowledge, not understanding the appraisal targets, bias against appraisees [45], bias on appraisal, misuse of authority, and use of their own criteria without paying attention to the organization's criteria [42]. In addition, some appraisers practice favoritism (i.e., mainly support their close employees), use their own high standards for assessing or judging the appraisees, and do not give feedback of the appraisal to the appraisees [41]. Problems with performance appraisal might also occur from the appraisees, because of their negative attitudes toward the appraisers, and not understanding the principles and targets of the appraisal [46]. Lavigne [47] found that problems with performance appraisal often occur from both appraisers and appraisees who do not understand targets, objectives, and processes of the performance appraisal. This leads to a negative atmosphere and relationship between subordinates and supervisors.

According to empirical research, unfairness of appraisal has a significant influence on JS [17]. Similarly, Suliman and Al Kathairi [19] found that effective and fair performance appraisal led to higher OC. By contrast, perception of unfairness or ineffective performance appraisal significantly reduced OC. Isenhour, Stone, Lien, Zheng, Zhang, and Li [22] reviewed the latest research and tested the effect of performance appraisal on OCB. They found that effective performance appraisal had a significant influence on altruism, conscientiousness, sportsmanship, courtesy, and civic virtue. Isenhour, Stone, Lien, Zheng, Zhang and Li [22] studied performance appraisals and organizational citizenship behaviors, finding that organizations' effective performance appraisal had a statistically significant effect on behaviors of helping others, sense of duty, sportsmanship, consideration, and cooperation. Meanwhile, Ahmed et al. [48], Bauwens et al. [49], Chattopadhyay [50], Teh et al. [51], Lu et al. [52], and Zheng et al. [53] studied performance appraisals and citizenship behaviors to organizations in different contexts. The findings of these studies are consistent in that effective performance appraisals enhance employees' good citizenship to organizations, whereas ineffective performance appraisals (with appraisal problems in process and person) have effects on low organizational citizenship behaviors. Based on the above literature review, the first and second research hypotheses were postulated:

**Hypotheses 1.** *Problems with performance appraisal on the appraisal process and the person directly influence OCB.*

**Hypotheses 2.** *Problems with performance appraisal on the appraisal process and the person indirectly influence OCB through JS and OC.*

*2.3. Job Satisfaction as a Mediator*

JS is an important variable in studies of work attitudes. Greenberg and ve Baron [54] proposed that JS influences work interaction. Each person may have a positive or negative attitude towards work. Congruently, Hackman and Oldham [55] stated that JS refers to the extent of employees' happiness with their work responsibility. In a meta-analysis, Whitman et al. [56] found that JS had a significant influence on OCB. Similarly, other meta-analyses [57,58] also revealed relationships between JS and OCB. Ilies et al. [59] demonstrated that JS had positive effects as a mediator of OCB, while Paillé [60] found that JS was a mediator in the form of employees' willingness, and such satisfaction had effects on OCB. Sendjaya et al. [61] studied JS as a mediator for predicting good behavior. They found that

employees' happiness and satisfaction with their jobs led to the high levels of positive behavior expected by organizations. Based on the above literature review, the third hypothesis was postulated:

**Hypotheses 3.** *JS is a mediator between problems with performance appraisal on the appraisal process and the person and OCB.*

*2.4. Organizational Commitment as a Mediator*

Steers and Porter [62] defined OC as the close and firm relationship of members in an organization. OC also refers to behavior congruent with values and a culture of integrity, and a willingness to devote physical and mental effort to participate in various company activities. OC has three aspects: (1) a strong belief and acceptance of targets; (2) a willingness to devote full effort; and (3) a strong desire to maintain membership [63].

Maharaj and Schlechter [64] found that OC had a positive relationship with OCB, while Allameh et al. [65] determined that levels of OC had variable relationships with OCB. Furthermore, Javadi and Yavarian [66] stated that indicators of positive OCB reflected employees' targets to devote maximum physical and mental efforts. Mowday [67] also found that OC was the main mediator between human resource management and work performance, while a meta-analytic study by Colquitt et al. [68] showed that fairness of work procedures and appraisals had positive effects on OCB through OC as a mediator. Congruently, Ahmed et al. [69] found that OC was a mediator between perception of fairness in performance appraisal and OCB. In other words, when employees perceive fairness in performance appraisal, they commit themselves to organizations and manifest good citizenship behavior. Based on the above literature review, the fourth research hypothesis was postulated:

**Hypotheses 4.** *Employees' organizational commitment is a mediator between problems with performance appraisal on the appraisal process and the person and OCB.*

Based on the perception of equity theory concerning how performance appraisal affects employees' positive behavior and empirical research evidence, JS and OC play roles as mediators between problems with performance appraisal and OCB. Accordingly, JS and OC were analyzed and synthesized in the study framework shown in Figure 1.

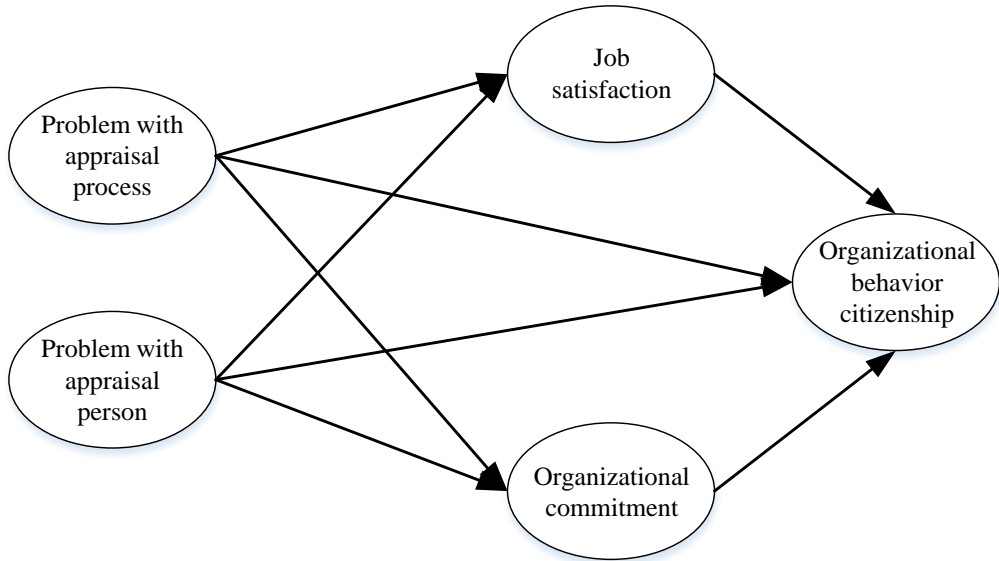

**Figure 1.** Conceptual research framework.

### 2.5. Research Objectives and Questions

Most studies concerning performance appraisal, JS, OC, and OCB focus on only two variables at a time, such as performance appraisal and JS [70], influence of performance appraisal on OC [30], or the relationship between performance appraisal and OCB [22]. As a result, data cannot be integrated to explain the different emerging phenomena. Moreover, empirical evidence from previous studies is scarce and inadequate to understand the different phenomena and enhance effective decision-making [71]. Therefore, in this study, several variables were examined to confirm their validity and reliability for explaining OCB. Three main research questions were postulated: (1) Is the perception model of problems with performance appraisal, JS, OC, and OCB congruent with the empirical evidence? (2) Are JS and OC mediators between the perception of problems with performance appraisal and OCB? and (3) How do the problems with performance appraisal influence OCB? Results will be useful for researchers, educators, students, human resource officers, and the general public who are interested in studying problems with performance appraisal.

## 3. Research Methodology

### 3.1. Population and Samples

The population of this study were 2735 employees in 12 auto parts manufacturers only at an operational level with responsibility for production of auto parts in Navanakorn Industrial Estate [72], located in the Eastern Region of Thailand. Sample size was determined according to the concept of Hair et al. [73] who proposed that for data analysis involving structural equation modeling, the sample size should be large, with 10 sample units for each parameter. In this study, there were 45 parameters, so the sample units should number 450. Samples were selected randomly with equal area representation using the simple random method. A total of 427 completed questionnaires were gathered.

### 3.2. Measurement, Validity, and Reliability

The scale for measuring problems of process and person performance appraisal was adapted from the employee perception scale of Na-Nan et al. [31]. A 16-item scale was developed, based on concepts and theories in performance appraisal, such as "the targets of the performance appraisal are not clear", "the performance appraisal lacks involvement between employees and administrators", "administrators do not give importance to real performance appraisal", and "the boss has bias and lacks fairness in performance appraisal". Cammann et al. [74] developed a questionnaire for measuring JS with three items modified from a scale for measuring satisfaction, known as the Michigan Organizational Assessment Questionnaire (OAQ); examples which include "all in all, I am satisfied with my job" and "in general, I like working here". Meyer and Herscovitch [75] developed a scale for measuring OC as a six-item questionnaire; for example, "working for organization's success is important for me", "I do not think to work for other organizations", and "I work according to the organization's operational guidelines", while Williams and Anderson [76] developed a scale for measuring OCB as a questionnaire with seven items, such as "assists supervisor with his/her work (when not asked)", "takes time to listen to co-workers' problems and worries", and "helps others who have heavy workloads".

In this study, a 16-item rating scale was selected with appropriate items using the 6-point Likert scale of strongly disagree (1), disagree (2), slightly disagree (3), slightly agree (4), agree (5), and strongly agree (6).

All the items were certified for content validity by five experts in organizational behavior, management, industrial psychology, human resource development, and testing and evaluation. Item validation results showed item-objective congruence (IOC) values between 0.40 and 1. Three items did not pass the set criteria, as two items had problems with performance appraisal and one item with organizational commitment. Therefore, these three items with scores less than 0.50 were removed, according to the criterion of Rovinelli and Hambleton [77] that appropriate IOC validation with five experts should be at least 0.80 to be considered as statistically significant.

The questionnaire quality was tested for reliability with Cronbach's alpha coefficient. JS had the highest internal consistency level (0.981) followed by OC, problems with person performance appraisal, OCB, and problems with process performance appraisal at 0.941, 0.893, 0.847, and 0.835, respectively. The internal consistency level of the whole questionnaire was determined at 0.908.

Convergent validity of the scale was tested according to the concept of Fornell and Larcker [78]. Confirmatory factor analysis was tested on the construct validity of each of the variable factors in the model to determine whether they were real factors, according to the concepts and theory with empirical evidence. Congruence was measured using Chi-square ($\chi^2$), relative Chi-square ($\chi^2/df$), goodness of fit index (GFI), adjusted goodness of fit index (AGFI), comparative fit index (CFI), standard root mean square residual (SRMR), and root mean square error of approximation (RMSEA) [79,80]. Table 1 illustrates the measurement of structural validity for each factor with a standardized factor loading. Each item or observable variable had large factor loadings (>0.50, significance at $p < 0.01$, all with the t statistic at more than 3). Therefore, all items had a significant relationship given the theoretical structure.

Composite reliability (CR) and average variance extracted (AVE) were calculated to test the construct reliability [78] of the scale and the structural model. As shown in Table 1, total confidence of each latent variable was between 0.860 and 0.916 (i.e., >0.7), indicating a good level of confidence. Regarding AVE, all latent variables were higher than 0.50 (i.e., >0.50). Therefore, all theoretical constructs were acceptable as psychological attributes.

Analysis of the mediators followed the concept of Baron and Kenny [81] using the AMOS Program with 5000 rounds of bootstrapping to examine whether they were real mediators, and to test the congruence between the theoretical and conceptual model and the empirical evidence.

**Table 1.** Confirmatory factor analysis of latent factors in the tested model.

| Latent Factors/Questions | Standardized Factor Loading | AVE and Composite Reliability ($\alpha$) |
|---|---|---|
| Problems with the appraisal process (Pro) | | |
| $\chi^2$ = 29.468, df = 21, *p*-value = 0.103, $\chi^2/df$ = 0.103, GFI = 0.985, AGFI = 0.967, RMR = 0.034, RMSEA = 0.031 | | |
| Pro | | |
| Eva1 | 0.779 | |
| Eva2 | 0.757 | |
| Eva3 | 0.815 | |
| Eva4 | 0.804 | $\alpha$ = 0.916 |
| Eva5 | 0.662 | AVE = 0.550 |
| Eva6 | 0.701 | |
| Eva7 | 0.819 | |
| Eva8 | 0.632 | |
| Eva9 | 0.682 | |
| Problems with the appraising person (Per) | | |
| $\chi^2$ = 5.134, df = 4, *p*-value = 0.274, $\chi^2/df$ = 1.283, GFI = 0.995, AGFI = 0.982, RMR = 0.017, RMSEA = 0.026 | | |
| Per | | |
| Eva11 | 0.822 | |
| Eva12 | 0.912 | $\alpha$ = 0.885 |
| Eva13 | 0.732 | AVE = 0.609 |
| Eva15 | 0.740 | |
| Eva18 | 0.674 | |
| Job satisfaction (JS) | | |
| $\chi^2$ = 0.524, df = 1, *p*-value = 0.471, $\chi^2/df$ = 0.521, GFI = 0.999, AGFI = 0.995, RMR = 0.013, RMSEA = 0.000 | | |
| JS | | |
| JS1 | 0.819 | $\alpha$ = 0.879 |
| JS2 | 0.812 | AVE = 0.708 |
| JS3 | 0.892 | |

**Table 1.** *Cont.*

| Latent Factors/Questions | Standardized Factor Loading | AVE and Composite Reliability ($\alpha$) |
|---|---|---|
| Organizational commitment (OC) | | |
| $\chi^2 = 4.616$, df = 3, *p*-value = 0.202, $\chi^2$/df = 1.539, GFI = 0.996, AGFI = 0.978, RMR = 0.015, RMSEA = 0.036 | | |
| OC | | |
| OC1 | 0.742 | |
| OC2 | 0.882 | $\alpha = 0.860$ |
| OC3 | 0.761 | AVE = 0.555 |
| OC4 | 0.642 | |
| OC5 | 0.673 | |
| Organizational citizenship behavior (OCB) | | |
| $\chi^2 = 8.847$, df = 8, *p*-value = 0.355, $\chi^2$/df = 1.106, GFI = 0.994, AGFI = 0.979, RMR = 0.018, RMSEA = 0.016 | | |
| OCB | | |
| OCB1 | 0.804 | |
| OCB2 | 0.812 | |
| OCB3 | 0.803 | $\alpha = 0.862$ |
| OCB4 | 0.667 | AVE = 0.515 |
| OCB5 | 0.616 | |
| OCB6 * | - | |
| OCB7 | 0.566 | |

Note: * Item deleted due to low factor loading, AVE refers average variance extracted, CR refers Composite reliability, Eva1–Eva9 indicate questions of problems with the appraisal process, Eva11–Eva18 indicate problems with the appraising person, JS1–JS3 indicate questions of job satisfaction, OC1–OC5 indicate question of organizational commitment, OCB1–OCB7 indicate questions of organizational citizenship behavior.

## 4. Data Analysis

More than half of the respondents (59%) were female and the rest (41%) were male. About a third were 31–35 years old (31%), followed by 26–30 (22%), and 40 years and above (13%). Regarding marital status, about half of the respondents were married (53%), followed by single (43%), and divorced/separated (4%). In terms of educational levels, about half of the respondents were graduates with a bachelor's degree (48%), followed by below a bachelor's degree (36%), and a doctoral degree (1%). About one-third of the respondents had work experience of 6–10 years (30%), followed by more than 10 years (23%), and less than 1 year (6%). About four-fifths of the respondents were employees (84%) and the others were senior employees (15.95%).

Regarding the studied variables, the factor of job satisfaction had the highest mean at 4.67 with the standard deviation at 0.87; followed by citizenship behaviors (mean 4.56, SD 0.78), employee engagement to organizations (mean 4.51, SD 0.87), problems of performance appraisals in process (mean 3.54, SD 1.11), and problems of performance appraisals in person (mean 3.24, SD 1.01).

The four-step test of Baron and Kenny [81] was used in four steps to determine whether the studied variables were real mediators or not as follows.

**Step 1:** Influence of the independent variable (X) on mediators (M) was assessed on the path of $X \overset{c}{\to} Y$ with statistical significance of c. A c value higher than 0.2 was regarded as abnormally high, indicating the existence of mediators. Test results showed that the independent variable, problems with performance appraisal, included two variables as problems with the appraisal process and problems with the appraising person. The influence coefficients on OCB ranged between −0.403 and −0.313, with a statistical significance level of <0.01, as represented in Scheme 1.

**Step 2:** Influence of the independent variable (X) on mediator (M) was assessed on the path of $X \overset{a}{\to} M$ with statistical significance of a. Results showed that problems with the appraisal process and problems with the appraising person as independent variables, influenced JS as the mediator with influence coefficients between −0.374 and −0.229, with a statistical significance level of <0.01, as represented in Scheme 2. Problems with the appraisal process and problems with the appraising person as independent variables influenced the factor of OC as the mediator, with influence coefficients between −0.460 and −0.244, with a statistical significance level of <0.01, as represented in Scheme 3.

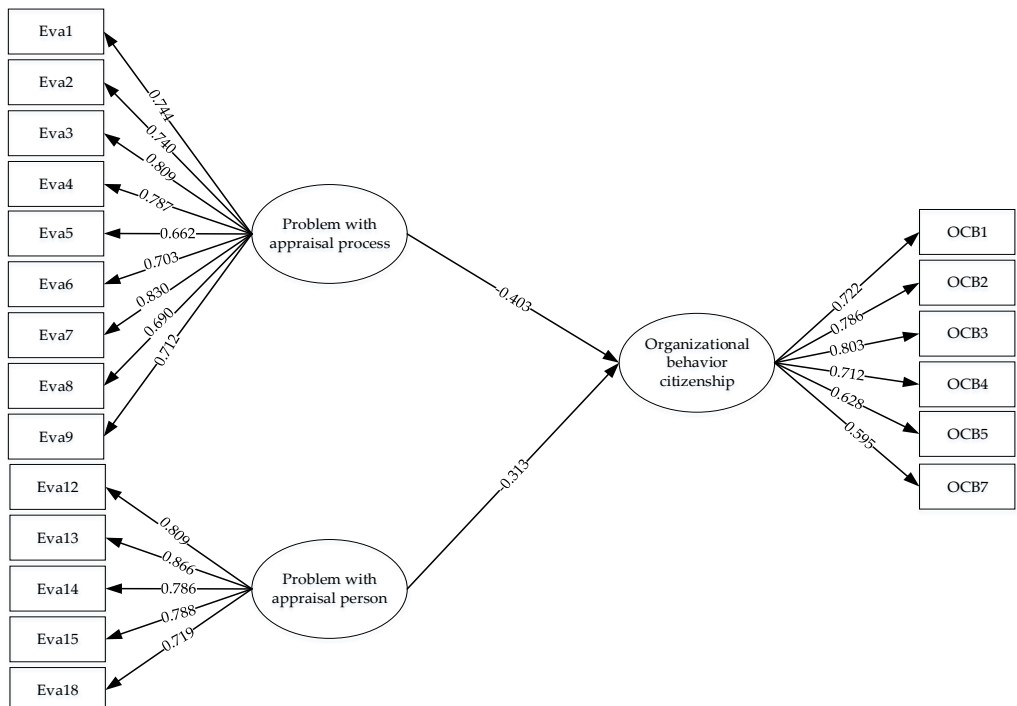

**Scheme 1.** Test of total influence.

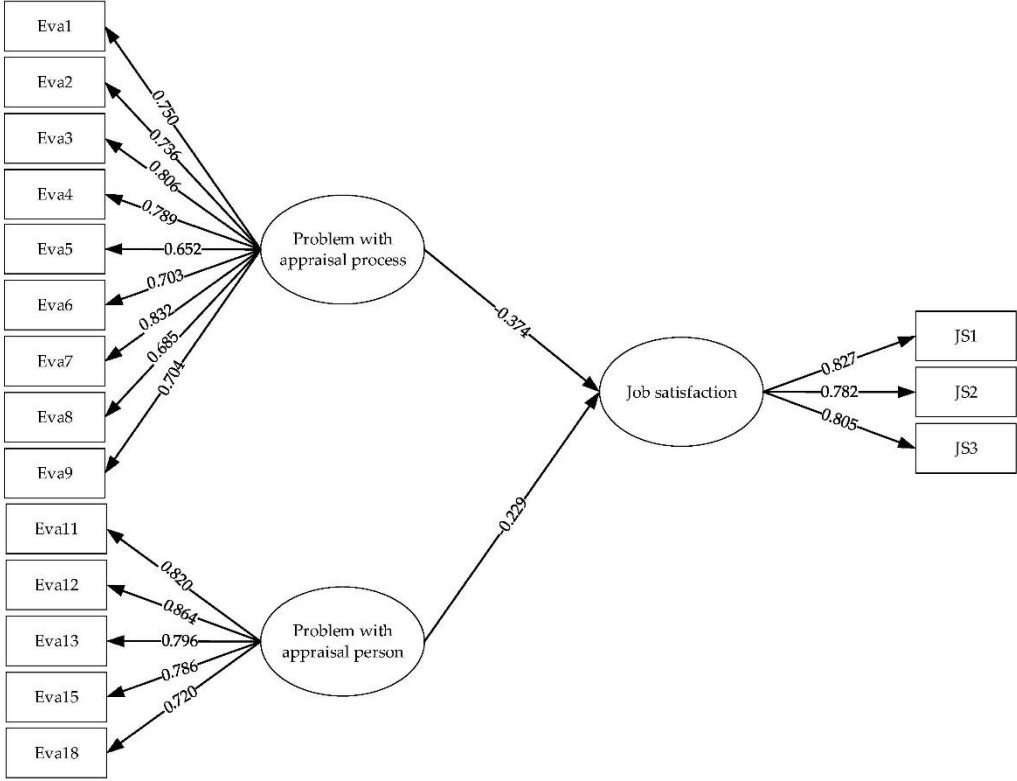

**Scheme 2.** Influence of appraisal problems on job satisfaction (JS).

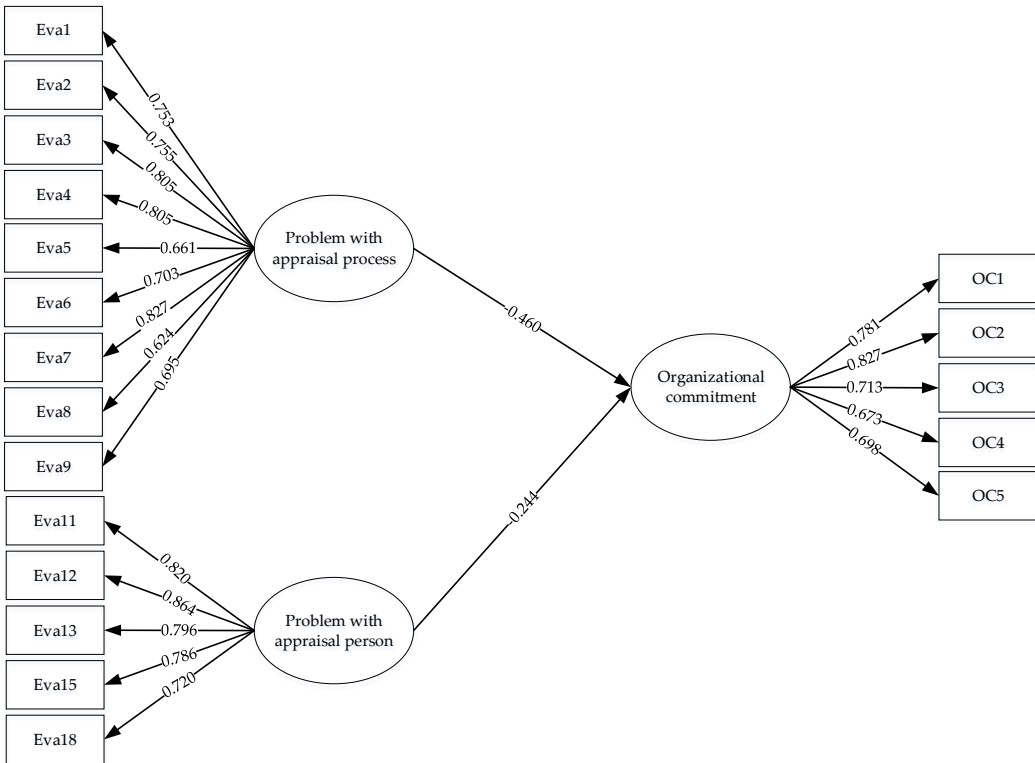

**Scheme 3.** Influence of appraisal problems on organizational commitment (OC).

**Step 3:** Influence of the mediators (M) on the dependent variable (Y) was assessed on the path of M $\overset{b}{\to}$ Y with statistical significance of b. Test results showed that OC and JS as the mediators influenced the factor of OCB as the dependent variable, with influence coefficients between 0.662 and 0.199, with a statistical significance level of <0.01, as represented in Scheme 4.

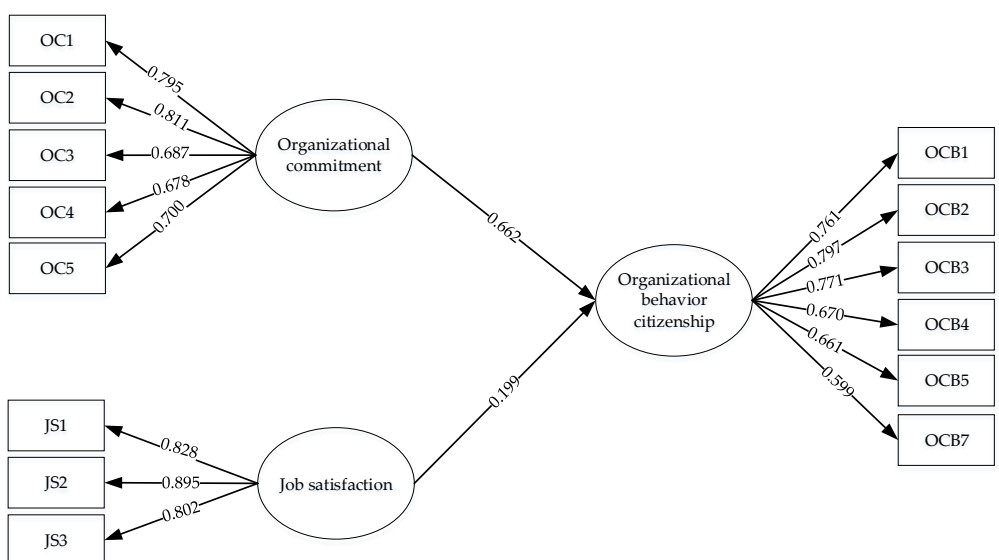

**Scheme 4.** Influence of organizational commitment and job satisfaction on organizational citizenship behavior (OCB).

Model analyses of Schemes 1–4 found that the concepts and theory were congruent with the empirical evidence when considering most values of model fit indices, as shown in Table 2.

**Table 2.** Model fit indices.

| Model | $\chi^2$ | df | *p* | CFI | GFI | AGFI | TLI | PGFI | RMR | RMSEA |
|---|---|---|---|---|---|---|---|---|---|---|---|
| 1a | 167.308 | 140 | 0.058 | 0.994 | 0.964 | AGFI | 0.992 | 0.642 | 0.021 | 0.058 |
| 1b | 148.016 | 123 | 0.062 | 0.995 | 0.965 | AGFI | 0.993 | 0.625 | 0.052 | 0.022 |
| 1c | 119.849 | 98 | 0.066 | 0.993 | 0.967 | 0.948 | 0.995 | 0.619 | 0.041 | 0.023 |
| 1d | 73.934 | 60 | 0.107 | 0.996 | 0.967 | 0.958 | 0.994 | 0.642 | 0.058 | 0.021 |

Remarks: CFI—comparative fit index, GFI—goodness of fit index, AGFI—adjusted goodness of fit index, TLI—Tucker Lewis index, PGFI—parsimonious goodness of fit, RMSEA—root mean square error of approximation and RMR—root mean square residual.

**Step 4:** Full model analysis considered that c' should be less than c, resulting from additional M as the mediator between the independent and dependent variables. However, Preacher and Hayes [82] stated that in model analysis with mediators, the influence of c should not be significant. If the influence of c' is statistically significant, the mediator is regarded as a partial mediator. By contrast, if c' is not statistically significant, the mediator is regarded as a full mediator. In this study, results showed that c' (problems with the appraisal process and problems with the appraising person) reduced values without statistical significance (i.e., 0.087 and 0.020), whereas problems with the appraisal process influenced JS and OC at −0.342 and −0.447, with a significance level of <0.01, and problems with the appraising person influenced JS and OC at −0.218 and −0.231, with a significance level of <0.01. These results indicated that the mediators were full mediators.

Analysis of the structural equation, as illustrated in Figure 2, revealed that the model influence of perception on problems with performance appraisal, JS, OC, and OCB was congruent with the empirical evidence. The indexes were $\chi^2$ = 329.891, df = 294, *p* = 0.073, $\chi^2$/df = 1.122, CFI = 995, GFI = 0.949, AGFI = 0.920, TLI = 994, PGFI = 0.687, RMSEA = 0.017, and RMR = 0.055. All indices conformed to the set criteria, indicating that the model developed from concepts and theory fitted well with the empirical evidence.

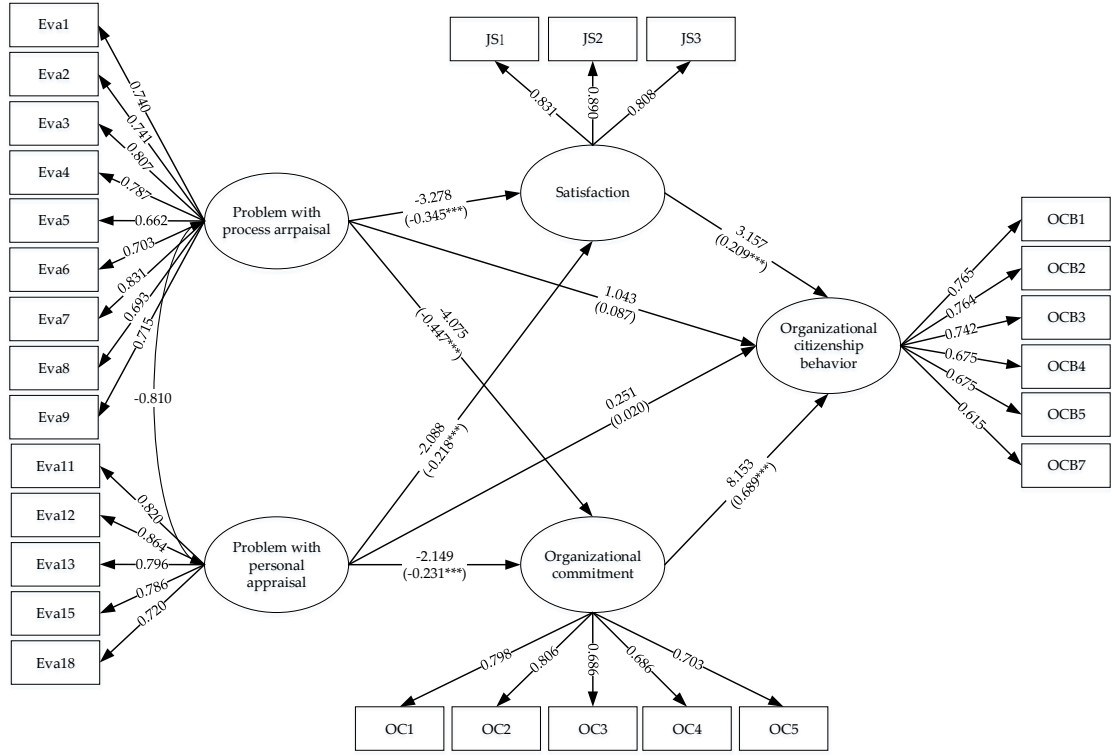

**Figure 2.** Adjusted structural equation according to program suggestions and analysis of 5000 rounds of bootstrapping. *** means significant level at 0.001.

The first research question was whether the perception model of problems with performance appraisal and OCB, with JS and OC as mediators, was congruent with the empirical evidence. Analysis results indicated that the model developed from concepts and theory was congruent with the empirical evidence, since all fit indices satisfied the criteria. In other words, the model developed on the basis of concepts and theories concerning problems with performance appraisal explained OCB with the effective mediators of JS and OC.

Mediator variables were tested by the resampling method with replacement of 5000 sets. In this study there were 427 sample units, so sampling with replacement might obtain repetitive units and these were not regarded as a mistake. Each set of data was used for the regression analysis to identify dependent, independent, and mediator variables. Analysis results obtained the path coefficient along the path to and from the mediators, and the standard error (SE) of each of the 5000 sets. Then, both types were presented, as shown in Table 3.

**Table 3.** Results of 5000 rounds of bootstrapping analysis.

| Parameter | | | Estimate | SE | $t$ | 95% CI | | $p$ |
|---|---|---|---|---|---|---|---|---|
| | | | | | | Lower | Upper | |
| JS | <— | Per | −0.281 | 0.086 | −3.278 | −0.450 | −0.131 | <0.01 |
| OC | <— | Per | −0.322 | 0.079 | −4.075 | −0.477 | −0.179 | <0.01 |
| JS | <— | Pro | −0.185 | 0.089 | −2.088 | −0.353 | −0.033 | <0.05 |
| OC | <— | Pro | −0.174 | 0.081 | −2.149 | −0.323 | −0.027 | <0.05 |
| OCB | <— | JS | 0.196 | 0.062 | 3.157 | 0.054 | 0.353 | <0.05 |
| OCB | <— | CC | 0.727 | 0.089 | 8.153 | 0.542 | 0.944 | <0.01 |
| OCB | <— | Per | 0.066 | 0.063 | 1.043 | −0.035 | 0.168 | 0.304 |
| OCB | <— | Pro | 0.016 | 0.065 | 0.251 | −0.082 | 0.129 | 0.780 |

Type 1 was calculated for the average product of path coefficients and the average SE, and then for the t statistic and the significant level if |t| > 2.00 to show a significance level at 0.05.

Type 2 used the path coefficient along the path to and from the mediators with 5000 values ranked in ascending order. Intervals of these values were determined for coverage at percentiles from 2.5 to 97.5. Coverage of zero indicates that the product is not different from zero at a significance level of 5% [83]. Results of mediator analysis showed that the variables of job satisfaction and organizational commitment ranged from percentile 2.5 to percentile 97.5, without coverage of zero. This indicated that both variables were real mediators between problems with performance appraisal on process and person, as shown in Table 2.

According to Table 2, JS and OC are latent factors that might relate to variables of problems with performance appraisal by process and person to OCB. Results showed that when the variables of JS and OC were added as mediators among the problems of performance appraisal on process and person together with OCB, the path coefficients reduced considerably. The significance test of mediator influence was supported with empirical evidence and showed that the two indirect influence paths were statistically significant ($p < 0.05$). This indicated attributes of the factors of problems with performance appraisal by person and process. This conformed to Hypothesis 3 and 4. Although it has an important role in OCB, JS should occur before OCB. Similarly, employees should obtain fairness in performance appraisal, both for process and person, without problems before manifesting OCB.

The second research question was whether JS and OC were mediators between perception of problems with performance appraisal and OCB. Analysis results suggested that JS and OC were real mediators between perception of problems with performance appraisal for the process and the person and OCB.

According to Table 4, the perception variables of problems with performance appraisal on process had no statistically significant direct influence on OCB (β = 0.087 and −0.205). Thus, Hypothesis 1 was not confirmed. Meanwhile, results showed that the perception variables of problems with performance appraisal on process influenced OCB through JS and OC with statistical significance

at <0.01 (β = −0.293 and −0.185). This finding confirmed Hypothesis 2. Perception variables of problems with performance appraisal on process had direct influence on JS and OC with a statistical significance level of <0.05 (β = 0.347 and −0.446). On the other hand, perception variables of problems with performance appraisal on person had direct influence on factors of JS and OC with a statistical significance level of <0.05 (β = 0.220 and −0.230), whereas factors of JS and OC still had direct influence on factors of OCB, with a statistical significance level of <0.05.

**Table 4.** Factors with direct, indirect, and overall influence.

| Variable | JS | | | OC | | | OCB | | |
|---|---|---|---|---|---|---|---|---|---|
| | DE | IE | TE | DE | IE | TE | DE | IE | TE |
| Pro | −0.347 | - | −0.347 | −0.446 | - | −0.446 | 0.087 | −0.380 | −0.293 |
| Per | −0.220 | - | −0.220 | −0.230 | - | −0.230 | 0.020 | −0.205 | −0.185 |
| JS | - | - | - | - | - | - | 0.214 | - | 0.214 |
| OC | - | - | - | - | - | - | 0.686 | - | 0.686 |

Remarks: DE—direct effect, IE—indirect effect, TE—total effect.

Regarding the squared multiple correlation coefficient (SMC) or predicting coefficient ($R^2$), the factor of JS had a value that explained the variance of JS at 4.5% and the factor of OC explained the variance of OC at 8.6%. This indicated that problems of performance appraisal on person and on process gave low JS and OC. The four studied parameters for predicting the variable of OCB explained 70% of the variance of OCB, indicating that problems with performance appraisal on person and on process result in very low JS and OCB. If organizations improve methods to build JS and OC, they will increase OCB to a high level.

The data presented answered the last research question concerning how the perception of problems with performance appraisal influenced OCB. Results indicated that problems with performance appraisal (on the process and on the person) did not influence OCB directly when there were mediators. In other words, problems with performance appraisal (on the process and on the person) influenced OCB through JS and OC as the mediators.

## 5. Discussion

The causal relationship model of perception on problems with performance appraisal and OCB through mediators of JS and OC was congruent with the empirical evidence. Model fit index values were $\chi^2 = 329.891$, df = 294, $p = 0.073$, $\chi^2/\text{df} = 1.122$, GFI = 0.949, AGFI = 0.920, PGFI = 0.687, RMSEA = 0.017, and RMR = 0.055. All values passed the criteria with $p$-values of more than 0.05 [84]. The value of $\chi^2/\text{df}$ was between 2.00 and 5.00 [85], RMR should not be more than ±2.58 [86], RMSEA should be less than 0.05, GFI should be between 0.90 and 0.95 [84], AGFI should be more than 0.95 [87], and PGFI should be more than 0.50 [88].

The data were statistically tested to examine whether the variables of JS and OC as mediators were congruent with the concepts and theories. Results showed that both variables were mediators of the influences of perceived problems with performance appraisal on OCB. Findings confirmed the results of Paillé [60] and Sendjaya, Pekerti, Cooper, and Zhu [61] that JS was a mediator for OCB. In other words, if companies desire their employees to manifest OCB, they need to build JS as a good sense of duty. Work staff will then operate at full potential and accept company guidelines on behavior. Meanwhile, OC was also found as a mediator in this study. Mowday [67], Colquitt, Conlon, Wesson, Porter, and Ng [68], and Ahmed, Ramzan, Mohammad, and Islam [69] also found OC as a mediator when accepting targets, devoting effort to work, and desiring to work harder to reflect good organizational behavior. If employees highly commit to an organization, this will lead to effectiveness of work operation or OCB.

In the causal model that was tested in this study, problems with performance appraisal influenced OCB through the mediators of JS and OC. Our findings were consistent with the concepts and theories

of previous studies. This model will be useful for educators or researchers who want to study and explain the phenomenon of OCB. This model includes the variables of problems with performance appraisal on the process and on the person that have indirect influence on the variable of OCB through JS and OC. The findings confirmed that JS and OC were mediators affecting the prediction of OCB quite highly, by accounting for 70% of the variance. The data revealed the importance of these two variables in explaining OCB.

In addition, JS and OC will also contribute to the development of organizational open innovation [89], whereby open innovation will help the organization's use of inflows and outflows of knowledge to accelerate internal innovation and expand the markets for external use of innovation [90–92]. Therefore, the importance of performance appraisal affects JS, OC, and OCB, which is considered to be a result of a person's behavior. These factors inevitably affect the emergence of open innovation and lead to the prosperity of the organization [93,94].

## 6. Conclusions

This study's research results revealed that perception models of appraisal problems, job satisfaction, employee engagement to organizations, and citizenship behaviors are consistent with previous empirical evidence by considering the congruence indexes set by researchers and educators. Job satisfaction and employee engagement to organizations are mediators between perception of appraisal problems and citizenship behaviors by analyzing mediators according to the concept of Baron and Kenny [1]. In addition, structural equation modelling was used with bootstrapping of 5000 rounds for the analysis. The results of the study showed that both variables conformed to the set conditions to test the given mediators. It was concluded that job satisfaction and employee engagement to an organization were important mediators between perceived problems of performance appraisals and citizenship behaviors. Regarding the effects of the variables, appraisal problems in the process and in the person had positive effects on citizenship behaviors directly and indirectly through job satisfaction and employee engagement to organizations. Finally, a revision of Adams' theory [24] of justice was proposed in this article.

### 6.1. Theoretical Implications

This study's findings revealed that problems with performance appraisal on both the process and on the person have an inverse influence on JS and OC and OCB. This result may be explained by the equity theory, which states that, when people are treated unfairly, their JS reduces and they do not commit to their organizations and do not manifest OCB. However, the influence of JS and OC on OCB can also be explained using the exchange theory. If organizations or administrators ensure that their employees are satisfied with their job, then positive OCB will result.

### 6.2. Practical Implications

Administrators and human resource managers may use the findings of this study to effectively explain the phenomenon of OCB. Results indicate that unfairness of performance appraisal leads to problems with the process and the appraiser that have negative effects on OCB, JS, and OC. Therefore, administrators should design the process of performance appraisal with fairness, transparency, and testability. They should also determine concrete indicators, a proper period of appraisal time, and use an appraisal form suitable for the employees' duties. In addition, both appraisers and appraisees should be educated through development and training about targets, objectives, and unbiased ratings in order to increase the fairness of performance appraisal.

The results of the study point out that problems in performance appraisals had negative effects directly and indirectly on organizational citizenship behaviors. Therefore, administrators and human resource managers should pay attention to solve these problems by adjusting the system and designing the process to solve all encountered problems. In addition, development of personnel related to performance appraisals should be arranged for these people to know, understand, and be capable in

testing and rating work performance. These improvements will result in performance appraisals which enhance employees' job satisfaction, organizational engagement, and good citizenship behaviors.

The study's results demonstrated that JS and OC function as mediators concerning problems with performance appraisal and OCB of employees. Therefore, administrators and human resource units should enhance and support employees to love and commit to organizations by creating an atmosphere that facilitates devotion of physical and mental efforts. Moreover, procurement, promotion, and support may be improved by using incentives to enhance employees' JS and effective work operation.

Based on Adams' equity theory [23], the findings of Na-Nan et al. [31], and practical implications found by Gilliland and Langdon [37], Shrivastava, Jones, Selvarajah, and Van Gramberg [38], Umair, Javaid, Amir, and Luqman [39], and Pichler [40], combined with the latest findings of this research study, it is proposed that problems with performance appraisal can be divided into two categories, consisting of problems with the appraisal process and problems with the appraising person in the effect of JS, OC, and OCB.

### 6.3. Limitations

The research reported in this article has several limitations. The study samples were operational employees and this fact may limit the generalizability of the results. Future researchers should collect data from administrators together with employees to extend the scope of the generalizations [91]. Moreover, the sample group of employees worked only in the automobile parts manufacturing industry. This industry has particular aspects of work that are different from other industries. Contexts of the study should also be expanded to include diversity in occupations, languages, societies, and cultures to increase the rigidity of the developed model. Furthermore, this research focused on only three variables (i.e., problems with performance appraisal, JS, and OC) that affected OCB. There are other variables that may predict OCB, such as perceived self-ability, leadership, work motivation, work engagement, and organizational environment. These variables may be studied in future research to expand our body of knowledge in this important field.

**Author Contributions:** Conceptualization, K.N.-N.; Methodology, K.N.-N.; Software, S.K.; Validation, K.N.-N., S.K., J.J. and I.D.S.; Formal Analysis, K.N.-N., S.K., J.J. and I.D.S.; Investigation, K.N.-N. and S.K.; Resources, S.K.; Data Curation, K.N.-N., S.K., J.J. and I.D.S.; Writing-Original Draft Preparation, K.N.-N.; Writing-Review & Editing, K.N.-N., J.J. and I.D.S.; Visualization, K.N.-N., S.K., J.J. and I.D.S.; Supervision, J.J. and S.I.D.; Project Administration, K.N.-N.; Funding Acquisition, S.K. All authors have read and agreed to the published version of the manuscript.

**Funding:** This research was funded by Faculty of Business Administration, Rajamangala University of Technology Thanyaburi.

**Conflicts of Interest:** The authors declare no conflict of interest.

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
