# Peer review of "Mediating Effects of Job Satisfaction and Organizational Commitment between Problems with Performance Appraisal and Organizational Citizenship Behavior"

_2199-8531, doi:10.3390/joitmc6030064_

Round 1

Reviewer 1 Report

The issues discussed in the article should be considered as important and valid in contemporary management sciences. Especially the study of the factors influencing the OCB of employees is particularly important for modern organizations, and scientific verification of those factors fills the research gap in the field of HRM. The authors of the study proposed a new model that, in terms of a more comprehensive approach than the current literature proposals, explains the impact of appraisal performance on the OCB of employees. Research was carried out using structural equation modeling.   The article noted the following specific shortcomings:  

  1. The entry in the Introduction "Therefore, in this study, factors of OCB were studied to determine whether they were influential in particular changing contexts and situations to raise awareness for improving and developing appraisal effectiveness" can possibly be considered as an aim of the study, but it is not in line with the content of the article that does not relate to the appraisal effectiveness, but to the impact of problems with performance appraisal on the OCB.  
  2. I believe that the content of the article lacks an exhaustive deduction of hypotheses concerning the mediating effect of JS and OC in relation to problems with performance appraisal. In my opinion, the considerstions on the H1 hypothesis are also insufficient (based only on the results of the research by Isenhour et al. and in my opinion the described phenomenon were insufficiently explained). Therefore, the deficiencies refer to the improper embedding of the examined problem in the literature. Moreover, it has not been sufficiently explained why mediating variables were limited to JS and OC, and e.g. work motivation or work engagement we’re not considered.   
  3. It was not shown how the variables were measured, especially there is no indication of the measurement of the variable "problems with performance appraisal", and the reference made to the literature (Na-Nan et al.) does not, in my opinion, provide a sufficient description of the tested construct. The article does not specify the classification of the analyzed problems with performance appraisal. It is also unknown what item set was used to measure the JS. Descriptive statistics values ​​(ie, M, SD) for individual variables are missing.  
  4. The research was conducted among 427 employees who worked at the operational level in automobile parts manufacturing companies in the Navanakorn Industrial Estate (it is not known from how many organizations the employees came from, and the characteristics of the research sample were not given). This fact was not cited in the discussion, conclustion, or practical and theoretical implications: whether the authors believe that the obtained results allow for generalizations that apply to all organizations. If so, on what basis.  
  5. The structure of the study can be regarded as the correct from the point of view of research methodology, although my doubts are raised by:   - inclusion in the summary of specific parameters relating to the developed model;   - failure to indicate the research gap in Intruduction, as well as the correctly and clearly defined purpose of the article;   - lack of the description of the research sample in the research part;   - I would consider combining Theoretical and practical implications as well as Limitations of the study into one part called Conclusions. The conclusions must also contain a reference to a properly defined purpose of the study.   
  6. I have doubts about the relation of the construct "fairness of performance appraisal" to "problems with performance appraisal", especially since the authors write that "fair performance appraisal is an important factor that affects employees' JS, OC and OCB". Moreover, in point 3.2. concerning directly the problems with performance appraisal, the authors describe in detail the issue of fairness of performance appraisal.  
  7. In my opinion, H3 is contained in H2. It may be a mistake to put H2 in point 3.2. It would be more reasonable to include the hypothesis H2 in the part concerning OC in point 3.4.   
  8. In some places there is a missing footnote: "(Reference?)", "(Reference to our previous study)", or there is an incorrectly formatted footnote: "(Singh & Rana, 2013)" or "(Suliman & Al. Kathairi, 2012). "

Author Response

Dear Reviewers,

I am very appreciating that my manuscript has been reviewed and considered, thank you very much. All of the reviewer comments are very important to improve this paper appropriate for publication in “Journal of Open Innovation: Technology, Market, and Complexity — Open Access Journal”. Therefore, the 1st revised manuscript was carefully prepared and sent back for your consideration. Regarding to all questions, the corrections were defined in blue text with page number, please kindly find in the revision manuscript with tracked changes file. Any other additional point for the publication, please kindly again recommend. Lastly, I indeed hope that our correction would be satisfied and the paper will be published in the Journal of Open Innovation: Technology, Market, and Complexity — Open Access Journal. 

Associate Professor Ian David Smith, School of Education & Social Work, The University of Sydney, Sydney, NSW, AUS, proofread and editing manuscript before resubmission. 

Sincerely Yours’

The Corresponding author

Reviewer 1

The issues discussed in the article should be considered as important and valid in contemporary management sciences. Especially the study of the factors influencing the OCB of employees is particularly important for modern organizations, and scientific verification of those factors fills the research gap in the field of HRM. The authors of the study proposed a new model that, in terms of a more comprehensive approach than the current literature proposals, explains the impact of appraisal performance on the OCB of employees. Research was carried out using structural equation modeling.   The article noted the following specific shortcomings:  

  1. The entry in the Introduction "Therefore, in this study, factors of OCB were studied to determine whether they were influential in particular changing contexts and situations to raise awareness for improving and developing appraisal effectiveness" can possibly be considered as an aim of the study, but it is not in line with the content of the article that does not relate to the appraisal effectiveness, but to the impact of problems with performance appraisal on the OCB.  

Action/Answer: Thank you so much for your kind suggestion. Revision has been made on page 2 and 3, line 55 to 87 as below:

Performance appraisal is complex in practice. Although related working units attempt to find ways to build clear indicators of work performance and behavior, problems often arise between employees and organizations [20] resulting in  job dissatisfaction, lack of OC, bad behavior and reduction of work performance [21]. Researchers in human resource management and development pay a lot of attention to problems of performance appraisals. The examples of the problems include appraisers’ lack of important facts, unclear standards for performance appraisals, appraisers’ inattention to importance of appraisals, appraisers’ unpreparedness for reviewing employee performance appraisals, appraisers’ dishonesty and insincerity in appraisals, appraisers’ lack of appraising skills, employees’ not being informed about their appraisal results, organizations’ lack of  appropriate systems for rewards and penalty to support appraisals, no discussion between appraisers and appraisees, and appraisers’ unclear appraisals. These factors cause appraisal problems which are mostly from person and appraisal process. As a result, such performance appraisals are ineffective and result in employees’ bad attitude to organizations. If these problems continue, employees will repeat such behaviors when they are promoted to higher positions as administrators and appraisers. This will become a vicious circle which causes employees’ job dissatisfaction [18], reduction of organizational engagement [19], reduction of organizational citizenship behaviors [22], and organizations’ inability to compete effectively with industrial rivals, because capable employees are discouraged and eventually leave their organizations.  

Because of these problems, the researchers were interested in studying direct effects of problem factors in performance appraisals on citizenship behaviors, and indirect effects through mediators of job satisfaction on factors of employees’ organizational engagement.

Research objectives and questions

Most studies concerning performance appraisal, JS, OC and OCB focus on only two variables at a time, such as performance appraisal and JS [23], influence of performance appraisal on OC (Singh & Rana, 2013) or the relationship between performance appraisal and OCB [22]. As a result, data cannot be integrated to explain the different emerging phenomena. Moreover, empirical evidence from previous studies is scarce and inadequate to understand the different phenomena and enhance effective decision-making [24]. Therefore, in this study, several variables were examined to confirm their validity and reliability for explaining OCB. Three main research questions were postulated: 1) Is the perception model of problems with performance appraisal, JS, OC, and OCB congruent with the empirical evidence?;  2) Are JS and OC mediators between the perception of problems with performance appraisal and OCB?; and 3) How do the problems with performance appraisal influence OCB? Results will be useful for researchers, educators, students, human resource officers and the general public who are interested in studying problems with performance appraisal.

  1. I believe that the content of the article lacks an exhaustive deduction of hypotheses concerning the mediating effect of JS and OC in relation to problems with performance appraisal. In my opinion, the considerstions on the H1 hypothesis are also insufficient (based only on the results of the research by Isenhour et al. and in my opinion the described phenomenon were insufficiently explained). Therefore, the deficiencies refer to the improper embedding of the examined problem in the literature.

Action/Answer: Thank you so much for your kind suggestion. Revision has been made on page 5, line 170 to 181 as below:

Isenhour, Stone, Lien, Zheng, Zhang and Li [22] studied performance appraisals and organizational citizenship behaviors, finding that organizations’ effective performance appraisal had a statistically significant effect on behaviors of helping others, sense of duty, sportsmanship, consideration, and cooperation. Meanwhile, the studies of Ahmed, et al. [49], Bauwens, et al. [50], Chattopadhyay [51], Teh, et al. [52], Lu, et al. [53], and Zheng, et al. [54] studied performance appraisals and citizenship behaviors to organizations in different contexts. The findings of these studies are consistent, that effective performance appraisals enhance employees’ good citizenship to organizations, whereas ineffective performance appraisals (with appraisal problems in process and person) have effects on low organizational citizenship behaviors. Based on the above literature review, the first and second research hypotheses were postulated:

            H1: Problems with performance appraisal on the appraisal process and the person direct influence OCB.

            H2: ….

Moreover, it has not been sufficiently explained why mediating variables were limited to JS and OC, and e.g. work motivation or work engagement we’re not considered.  

Action/Answer: Thank you so much for your kind suggestion. This study focused only on JS and OC variables as intermediate variables. However, the researchers will specify this fact in the limitations and suggestions for future research. Revision has been revised on page 16, line 496 to 500 as below:

Furthermore, this research focused on only three variables (problems with performance appraisal, JS and OC) that affected OCB. There are other variables that may predict OCB, such as perceived self-ability, leadership, work motivation, work engagement or organizational environment. These variables may be studied in future research to expand our body of knowledge in this important field.

  1. It was not shown how the variables were measured, especially there is no indication of the measurement of the variable "problems with performance appraisal", and the reference made to the literature (Na-Nan et al.) does not, in my opinion, provide a sufficient description of the tested construct.

Action/Answer: Thank you so much for your kind suggestion. Revision has been made on page 6 and 7, line 239 to 256 as below:

The scale for measuring problems of process and person performance appraisal was adapted from the employee perception scale of Na-Nan, et al. [73]. A 16-item scale was developed, based on concepts and theories in performance appraisal, such as “the targets of the performance appraisal are not clear”, “the performance appraisal lacks involvement between employees and administrators”, “administrators do not give importance to real performance appraisal”, and “the boss has bias and lacks fairness in performance appraisal”. Cammann, et al. [74] developed a questionnaire for measuring JS with three items modified from a scale for measuring satisfaction, known as the Michigan Organizational Assessment Questionnaire (OAQ). Item examples include “all in all, I am satisfied with my job”, and “in general, I like working here”. Meyer and Herscovitch [75] developed a scale for measuring OC as a six-item questionnaire for example “Working for organization’s success is important for me”, “I do not think to work for other organizations”, and “I work according to the organization’s operational guidelines” while Williams and Anderson [76] developed a scale for measuring OCB as a questionnaire with seven items such as “assists supervisor with his/her work (when not asked)” “takes time to listen to co-workers’ problems and worries”, and helps others who have heavy work loads”.

In this study, a 16-item instrument was selected with appropriate items using the 6-point Likert scale of strongly disagree (1), disagree (2), slightly disagree (3), slightly agree (4), agree (5) and strongly agree (6).

The article does not specify the classification of the analyzed problems with performance appraisal. It is also unknown what item set was used to measure the JS. Descriptive statistics values ​​(ie, M, SD) for individual variables are missing.  

Action/Answer: Thank you so much for your kind suggestion. Revision has been made on page 9, line 300 to 303 as below:

Regarding the studied variables, the factor of job satisfaction had the highest mean at 4.67 with the standard deviation at .87; followed by citizenship behaviors (mean 4.56, SD .78), employee engagement to organizations (mean 4.51, SD .87), problems of performance appraisals in process (mean 3.54, SD 1.11), and problems of performance appraisals in person (mean 3.24, SD .1.01). (Do you have statistical evidence that these rankings are statistically significant?)

  1. The research was conducted among 427 employees who worked at the operational level in automobile parts manufacturing companies in the Navanakorn Industrial Estate (it is not known from how many organizations the employees came from, and the characteristics of the research sample were not given).

Action/Answer: Thank you so much for your kind suggestion. Revision has been made on page 6, line 229 to 231 as below:

The population of this study was 2,735 employees in 12 auto parts manufacturers only at an operational level with responsibility for production of auto parts in Navanakhorn Industrial Estate [71]

This fact was not cited in the discussion, conclustion, or practical and theoretical implications: whether the authors believe that the obtained results allow for generalizations that apply to all organizations. If so, on what basis.  

Action/Answer: Thank you so much for your kind suggestion. Revision has been made on page 15 and 16, line 455 to 489 as below:

In the causal model that was tested in this study, problems with performance appraisal influenced OCB through the mediators of JS and OC. Our findings were consistent with the concepts and theories of previous studies. This model will be useful for educators or researchers who want to study and explain the phenomenon of OCB. This model includes the variables of problems with performance appraisal on the process and on the person that have indirect influence on the variable of OCB through JS and OC. The findings confirmed that JS and OC were mediators affecting the prediction of OCB quite highly, by accounting for 70% of the variance. The data revealed the importance of these two variables in explaining OCB.

Findings revealed that problems with performance appraisal on both the process and on the person have an inverse influence on JS and OC and OCB. This result can be explained by the equity theory. This states that, when people are treated unfairly, their JS reduces and they do not commit to organizations and do not manifest OCB. However, the influence of JS and OC on OCB can also be explained using the exchange theory. If organizations or administrators ensure that their employees are satisfied with their job, then positive OCB will result.

Administrators and human resource managers may use the findings of this study to effectively explain the phenomenon of OCB. Results indicate that unfairness of performance appraisal leads to problems with the process and the appraiser that have negative effects on OCB, JS and OC. Therefore, administrators should design the process of performance appraisal with fairness, transparency, and testability. They should also determine concrete indicators, a proper period of appraisal time, and use an appraisal form suitable for the employees’ duties. In addition, both appraisers and appraisees should be educated through development and training about targets, objectives, and unbiased ratings in order to increase the fairness of performance appraisal.

The results of the study demonstrate that problems in performance appraisals had negative effects directly and indirectly on organizational citizenship behaviors. Therefore, administrators and human resource departments should pay attention to solve these problems by adjusting the system and designing the process to solve all encountered problems. In addition, development of personnel related to performance appraisals should be arranged for these people to know, understand and be capable in testing and rating work performance. This is for performance appraisals to enhance employees’ job satisfaction, organizational engagement, and good citizenship behaviors.

The study results show that JS and OC function as mediators concerning problems with performance appraisal and OCB of employees. Therefore, administrators and human resource units should enhance and support employees to love and commit to their organizations by creating an atmosphere that facilitates devotion of their physical and mental efforts. Moreover, procurement, promotion and support may be improved by using incentives to enhance employees’ JS and effective work operation.

  1. The structure of the study can be regarded as the correct from the point of view of research methodology, although my doubts are raised by:  

- failure to indicate the research gap in Intruduction, as well as the correctly and clearly defined purpose of the article;  

Action/Answer: Thank you so much for your kind suggestion. Revision has been made on page 2, line 55 to 70 as below:

Performance appraisal is complex in practice. Although related working units attempt to find ways to build clear indicators of work performance and behavior, problems often arise between employees and organizations [20] resulting in  job dissatisfaction, lack of OC, bad behavior and reduction of work performance [21]. Researchers in human resource management and development pay a lot of attention to problems of performance appraisals. The examples of the problems include appraisers’ lack of important facts, unclear standards for performance appraisals, appraisers’ inattention to importance of appraisals, appraisers’ unpreparedness for reviewing employee performance appraisals, appraisers’ dishonesty and insincerity in appraisals, appraisers’ lack of appraising skills, employees’ not being informed about their appraisal results, organizations’ lack of  appropriate systems for rewards and penalty to support appraisals, no discussion between appraisers and appraisees, and appraisers’ unclear appraisals. These factors cause appraisal problems which are mostly from the person and appraisal process. As a result, such performance appraisals are ineffective and result in employees’ bad attitude to organizations. If these problems continue, employees will repeat such behaviors when they are promoted to higher positions as administrators and appraisers. This will become a vicious circle which cause employees’ job dissatisfaction [18], reduction of organizational engagement [19], reduction of organizational citizenship behaviors [22], and organizations’ inability to compete effectively with industrial rivals because good capable employees are discouraged and eventually leave their organizations.  

- lack of the description of the research sample in the research part;  

Action/Answer: Thank you so much for your kind suggestion. Revision has been made on page 6, line 229 to 231 as below:

The population of this study was a total of 2,735 employees in 12 auto parts manufacturers only at an operational level with responsibility for production of auto parts in Navanakhorn Industrial Estate [71]

 - I would consider combining Theoretical and practical implications as well as Limitations of the study into one part called Conclusions.

Action/Answer: Thank you so much for your kind suggestion. Revision has been made on page 15 and 16, line 464 to 502 as below:

            Findings revealed that problems with performance appraisal on both the process and on the person have an inverse influence on JS and OC and OCB. This result can be explained by the equity theory. This states that, when people are treated unfairly, their JS reduces and they do not commit to organizations and do not manifest OCB. However, the influence of JS and OC on OCB can also be explained using the exchange theory. If organizations or administrators ensure that their employees are satisfied with their job, then positive OCB will result.

Administrators and human resource managers may use the findings of this study to effectively explain the phenomenon of OCB. Results indicated that unfairness of performance appraisal leads to problems with the process and the appraiser that have negative effects on OCB, JS and OC. Therefore, administrators should design the process of performance appraisal with fairness, transparency, and testability. They should also determine concrete indicators, a proper period of appraisal time, and use an appraisal form suitable for the employees’ duties. In addition, both appraisers and appraisees should be educated through development and training about targets, objectives, and unbiased ratings in order to increase the fairness of performance appraisal.

The results of the study demonstrated that problems in performance appraisals had negative effects directly and indirectly on organizational citizenship behaviors. Therefore, administrators and human resource departments should pay attention to solve these problems by adjusting the system and designing the process to solve all encountered problems. In addition, development of personnel related to performance appraisals should be arranged for these people to know, understand and be capable in testing and rating work performance. This is for performance appraisals to enhance employees’ job satisfaction, organizational engagement, and good citizenship behaviors.

The study’s results showed that JS and OC function as mediators concerning problems with performance appraisal and OCB of employees. Therefore, administrators and human resource units should enhance and support employees to love and commit to organizations by creating an atmosphere that facilitates devotion of physical and mental efforts. Moreover, procurement, promotion and support may be improved by using incentives to enhance employees’ JS and effective work operation.

The research reported in this article has several limitations. The study samples were operational employees and this fact may limit the generalizability of the results. Future researchers should collect data from administrators together with employees to extend the scope of the generalizations. Moreover, the sample group of employees worked only in the automobile parts manufacturing industry. This industry has particular aspects of work that are different from other industries. Contexts of the study should also be expanded to include diversity in occupations, languages, societies, and cultures to increase the rigidness of the developed model. Furthermore, this research focused on only three variables (problems with performance appraisal, JS and OC) that affected OCB. There are other variables that may predict OCB, such as perceived self-ability, leadership, work motivation, work engagement or organizational environment. These variables may be studied in future research to expand our body of knowledge in this important field.

The conclusions must also contain a reference to a properly defined purpose of the study.  

Action/Answer: Thank you so much for your kind suggestion. Revision has been made on page 16 and 17, line 505 to 516 as below:

Research results revealed that perception models of appraisal problems, job satisfaction, employee engagement to organizations, and citizenship behaviors are consistent well to empirical evidence by considering from congruence indexes set by researchers and educators. Job satisfaction and employee engagement to organizations are mediators between perception of appraisal problems and citizenship behaviors by analyzing mediators according to the concept of Baron and Kenny [1]. In addition, structural equation modelling was used with bootstrapping of 5,000 rounds for the analysis. The results of the study showed that both variables conformed to the set conditions to test the given mediators. It was concluded that job satisfaction and employee engagement to organization were real mediators between perceived problems of performance appraisals and citizenship behaviors. Regarding the effects of the variables, appraisal problems (in process and in person) had positive effects on citizenship behaviors directly and indirectly through job satisfaction and employee engagement to their organizations.

  1. I have doubts about the relation of the construct "fairness of performance appraisal" to "problems with performance appraisal", especially since the authors write that "fair performance appraisal is an important factor that affects employees' JS, OC and OCB".

Action/Answer: Thank you so much for your kind suggestion. Revision has been made on page 4, line 137 to 141 as below:

The concepts and theory of justice can be used to explain performance appraisals, since the performance appraisals are based on justice of the processes designed by organizations and people who perform as appraisers, leading to dissatisfaction and negative results on employees’ behaviors and organizations. Therefore, appraisal problems are related to justice in the appraisal process and to the people who appraise performance. 

Moreover, in point 3.2. concerning directly the problems with performance appraisal, the authors describe in detail the issue of fairness of performance appraisal.  

Action/Answer: Thank you so much for your kind suggestion. Revision has been made on page 5, line 170 to 181 as below:

Isenhour, Stone, Lien, Zheng, Zhang and Li [22] studied performance appraisals and organizational citizenship behaviors, finding that organizations’ effective performance appraisal had a statistically significant effect on behaviors of helping others, sense of duty, sportsmanship, consideration, and cooperation. Meanwhile, the studies of Ahmed, et al. [49], Bauwens, et al. [50], Chattopadhyay [51], Teh, et al. [52], Lu, et al. [53], and Zheng, et al. [54] studied performance appraisals and citizenship behaviors to organizations in different contexts. The findings of these studies are consistent, that effective performance appraisals enhance employees’ good citizenship to organizations, whereas ineffective performance appraisals (with appraisal problems in process and person) have effects on low organizational citizenship behaviors. Based on the above literature review, the first and second research hypotheses were postulated:

H1: Problems with performance appraisal on the appraisal process and the person direct influence OCB.

H2: …

  1. In my opinion, H3 is contained in H2. It may be a mistake to put H2 in point 3.2. It would be more reasonable to include the hypothesis H2 in the part concerning OC in point 3.4.   8. In some places there is a missing footnote: "(Reference?)", "(Reference to our previous study)", or there is an incorrectly formatted footnote: "(Singh & Rana, 2013)" or "(Suliman & Al. Kathairi, 2012). " 

Action/Answer: Thank you so much for your kind suggestion.

This study had two main objectives: 1. analyses of direct and indirect effects (H1 and H2), and tests of mediators’ effect (H3 and H4) to find out whether they were full mediator or partial mediator. The researcher separated the hypotheses and analyses according to the principle of Baron and Kenny (1986), which suggest the method to analyze mediators.

Sorry for my mistake about writing references. (Singh & Rana, 2013)" and "(Suliman & Al. Kathairi, 2012) has been revised in MPDI style.

Reviewer 2 Report

Although the study deals with the relationship of interesting variables based on solid empirical studies, it is thought that there are many areas that need to be revised to be published.

1. Introduction

Interesting background and purpose of research should be presented to draw interest from readers. I don't feel that interest after reading the current introduction. Why should this study be carried out again, even though many similar studies have already been conducted? Please give me an interesting background of research to answer this question.

2. Research Objectives and questions

Same as above.

3. Theories for explaining the concepts and study framework

As the author suggests, there are three kinds of justice presented in the Justice literature. Why do the authors re-propose two kinds of problems with the PA, despite having such a great theory? What does this concept complement that existing justice theory does not explain? This is also linked to the theoretical contribution of this study. If it presents a better point than justice literature, then there is a clear theoretical contribution. But if not, it is not feasible why this concept was presented.

Another concern is that hypothesis 1, hypothesis 2, hypothesis 3, and hypothesis 4 can eventually be compressed into two hypotheses. One is that PA on the process is connected to OCB via JS and OC, and the other is that PA on the person is connected to OCB via JS and OC. The current hypothesis must be reorganized to prevent it from being repeated and fragmented.

4. Research methodology

In terms of methodology, there seems not much to be improved. However, during SEM analysis, CFI and TLI values are more appropriate criteria, so please report these values. Also, if the purpose of the CFA analysis is to ensure the discriminant validity between variables, make sure that the CFA should be conducted all items, not items for each variable.

5. Data analysis

By performing a single CFA with all variables considered, researchers can find the direct effect and indirect effect of each variable. Analyzing each separately is too repetitive. As the author suggests in the introduction, if the purpose of the study is to view the overall relationship (as seen in research model), report the CFA results taking into account the overall variables rather than the influence of individual variables.

6. Discussion

This part can be integrated with implications sections.

7/8. Theoretical implications / Practical implications

This part should be modified to provide more detailed and clear implications.

Author Response

Dear Reviewers,

I am very appreciating that my manuscript has been reviewed and considered, thank you very much. All of the reviewer comments are very important to improve this paper appropriate for publication in “Journal of Open Innovation: Technology, Market, and Complexity — Open Access Journal”. Therefore, the 1st revised manuscript was carefully prepared and sent back for your consideration. Regarding to all questions, the corrections were defined in blue text with page number, please kindly find in the revision manuscript with tracked changes file. Any other additional point for the publication, please kindly again recommend. Lastly, I indeed hope that our correction would be satisfied and the paper will be published in the Journal of Open Innovation: Technology, Market, and Complexity — Open Access Journal. 

Associate Professor Ian David Smith, School of Education & Social Work, The University of Sydney, Sydney, NSW, AUS, proofread and editing manuscript before resubmission. 

Sincerely Yours’

The Corresponding author

Reviewer 2

Although the study deals with the relationship of interesting variables based on solid empirical studies, it is thought that there are many areas that need to be revised to be published.

  1. Introduction

Interesting background and purpose of research should be presented to draw interest from readers. I don't feel that interest after reading the current introduction. Why should this study be carried out again, even though many similar studies have already been conducted? Please give me an interesting background of research to answer this question.

Action/Answer: Thank you so much for your kind suggestion. Revision has been made on page 2 and 3, line 55 to 73 as below:

Performance appraisal is complex in practice. Although related working units attempt to find ways to build clear indicators of work performance and behavior, problems often arise between employees and organizations [20] resulting in  job dissatisfaction, lack of OC, bad behavior and reduction of work performance [21]. Researchers in human resource management and development pay a lot of attention to problems of performance appraisals. The examples of the problems include appraisers’ lack of important facts, unclear standards for performance appraisals, appraisers’ inattention to importance of appraisals, appraisers’ unpreparedness for reviewing employee performance appraisals, appraisers’ dishonesty and insincerity in appraisals, appraisers’ lack of appraising skills, employees’ not being informed about their appraisal results, organizations’ lack of  appropriate systems for rewards and penalty to support appraisals, no discussion between appraisers and appraisees, and appraisers’ unclear appraisals. These factors cause appraisal problems which are mostly from person and appraisal process. As a result, such performance appraisals are ineffective and result in employees’ negative attitudes towards their organizations. If these problems continue, employees will repeat such behaviors when they are promoted to higher positions as administrators and appraisers. This will become a vicious circle which cause employees’ job dissatisfaction [18], reduction of organizational engagement [19], reduction of organizational citizenship behaviors [22], and organizations’ inability to compete effectively with industrial rivals, because capable employees are discouraged and eventually leave their organizations.  

Because of these problems, the researchers were interested in studying the direct effects of problem factors in performance appraisals on citizenship behaviors, and the indirect effects through mediators of job satisfaction on factors of employees’ organizational engagement.

  1. Research Objectives and questions

Same as above.

Action/Answer: Thank you so much for your kind suggestion. Revision has been made on page 2 and 3, line 75 to 87 as below:

Research objectives and questions

Most studies concerning performance appraisal, JS, OC and OCB focus on only two variables at a time, such as performance appraisal and JS [24], influence of performance appraisal on OC [25] or the relationship between performance appraisal and OCB [23]. As a result, data cannot be integrated to explain the different emerging phenomena. Moreover, empirical evidence from previous studies is scarce and inadequate to understand the different phenomena and enhance effective decision-making [26]. Therefore, in this study, several variables were examined to confirm their validity and reliability for explaining OCB. Three main research questions were postulated: 1) Is the perception model of problems with performance appraisal, JS, OC, and OCB congruent with the empirical evidence?;  2) Are JS and OC mediators between the perception of problems with performance appraisal and OCB?; and 3) How do the problems with performance appraisal influence OCB?. Results will be useful for researchers, educators, students, human resource officers and the general public who are interested in studying problems with performance appraisal.

  1. Theories for explaining the concepts and study framework

As the author suggests, there are three kinds of justice presented in the Justice literature. Why do the authors re-propose two kinds of problems with the PA, despite having such a great theory?

Action/Answer: Thank you so much for your kind suggestion. Revision has been made on page 4, line 137 to 163 as below:

The concepts and theory of justice can be used to clearly explain performance appraisals, since the performance appraisals are based on justice of the processes designed by organizations and people who perform as appraisers, leading to dissatisfaction and negative results on employees’ behaviors and organizations. Therefore, appraisal problems are related to justice in appraisal process and people who appraise performance. 

Appraisers often focus on explaining the emerging problems based on related concepts and theories. Accordingly, problems with performance appraisal can be classified into two main types: 1) Problems with the performance appraisal process include abstract appraisal indicators relying on appraisers’ judgment, lack of stakeholders’ involvement, old-fashioned and ineffective appraisal forms, unfairness of appraisal, inappropriate period of appraisal time, and discontinuity of the appraisal [43]. According to Paul [44], problems with the appraisal process include inappropriateness of appraisal criteria and data sources. Grund and Przemeck [45] studied problems with performance appraisal. They found that abstractness of the appraisal criteria or indicators may lead to appraisers’ bias on each appraisal, resulting in appraisees’ dissatisfaction. In some organizations, appraisal forms do not conform to the work contexts; they are so complicated that the appraisers are confused and do not understand the appraising methods, and the appraisal does not reflect the actual performance of the employees [46]. 2) Problems with the performance appraisal result from the appraisers’ lack of knowledge, not understanding the appraisal targets, bias against appraisees [47], bias on appraisal, misuse of authority and use of their own criteria without paying attention to the organization’s criteria [44]. In addition, some appraisers practice favoritism (i.e., mainly support their close employees), use their own high standards for assessing or judging the appraisees, and do not give feedback of the appraisal to the appraisees [43]. Problems with performance appraisal might also occur from the appraisees, because of their negative attitudes toward the appraisers, and not understanding the  principles and targets of the appraisal [48]. Lavigne [49] found that problems with performance appraisal often occur from both appraisers and appraisees who do not understand targets, objectives and processes of the performance appraisal. This leads to a negative atmosphere and relationship between subordinates and supervisors.

What does this concept complement that existing justice theory does not explain? This is also linked to the theoretical contribution of this study. If it presents a better point than justice literature, then there is a clear theoretical contribution. But if not, it is not feasible why this concept was presented.

Action/Answer: Thank you so much for your kind suggestion. Revision has been made on page 4, line 126 to 141 as below:

Equity theory (Adams [27] proposes that equity occurs when management is effective, transparent, and checkable, whereas inequality occurs when management is ineffective, not transparent, and unable to be checked. According to Gilliland and Langdon [39], fairness includes three aspects: 1) Procedural justice is fairness perceived by employees about the appropriateness of appraisal procedures in which the appraisees can express their opinions and receive feedback; the appraisal results are transparent and reliable, and the judgment procedure is not biased with double standards [40]; 2) Interpersonal fairness is people’s perception of the practice during the appraisal period, clear communication, honesty, ethics, and clear targets of performance appraisal [41]; and 3) Outcome fairness involves satisfaction of appraisal results that are comparable to work output. If the results are below employees’ expectations, a sense of unfairness will occur [42].

The concepts and theory of justice can be used to clearly explain performance appraisals, since the performance appraisals are based on justice of the processes designed by organizations and the people who perform as appraisers, leading to dissatisfaction and negative results on employees’ behaviors and organizations. Therefore, appraisal problems are related to justice in the appraisal process and the people who appraise performance. 

Another concern is that hypothesis 1, hypothesis 2, hypothesis 3, and hypothesis 4 can eventually be compressed into two hypotheses. One is that PA on the process is connected to OCB via JS and OC, and the other is that PA on the person is connected to OCB via JS and OC. The current hypothesis must be reorganized to prevent it from being repeated and fragmented.

Action/Answer: Thank you so much for your kind suggestion. Since this study had 2 main objectives to study direct and indirect effects with hypotheses of H1 and H2, and to test 2 mediators with H3 and H4.

The tests of mediators were according to the principle of Baron and Kenny (1986) which suggests how to test mediators by testing direct effects of independent variables on dependent variables. Then the mediators were tested to find out whether the variables were mediators. Therefore, the hypothesis testing was according to the above principle.

  1. Research methodology

In terms of methodology, there seems not much to be improved. However, during SEM analysis, CFI and TLI values are more appropriate criteria, so please report these values.

Action/Answer: Thank you so much for your kind suggestion. Revision has been made on page 12, line 342 and 346 as below:

Model

df

p

CFI

GFI

AGFI

TLI

PGFI

RMR

RMSEA

1a

167.308

140

0.058

0.994

0.964

AGFI

0.992

0.642

0.021

0.058

1b

148.016

123

0.062

0.995

0.965

AGFI

0.993

0.625

0.052

0.022

1c

119.849

98

0.066

0.993

0.967

0.948

0.995

0.619

0.041

0.023

1d

73.934

60

0.107

0.996

0.967

0.958

0.994

0.642

0.058

0.021

 = 329.891, df = 294, p = 0.073, /df = 1.122, CFI = 995, GFI = 0.949, AGFI = 0.920, TLI = 994,

PGFI = 0.687, RMSEA = 0.017, RMR = 0.055

  1. Data analysis

By performing a single CFA with all variables considered, researchers can find the direct effect and indirect effect of each variable. Analyzing each separately is too repetitive. As the author suggests in the introduction, if the purpose of the study is to view the overall relationship (as seen in research model), report the CFA results taking into account the overall variables rather than the influence of individual variables.

Action/Answer: Thank you so much for your kind suggestion. Regarding the objective to study direct and indirect effects, the researcher chose SEM analysis to consider the aspects of effects. Consistently, Hair, et al. (2017) suggest how to test the relationship of several independent variables which have direct effects on dependent variables and have effects through intervening variables to dependent variables on other paths, known as “indirect effect”. To analyze the relationships in order to answer the research questions, a set of mathematical equations are needed. Such set of mathematical equations is called structural equation modelling.

Hair, J. F., Sarstedt, M., Ringle, C. M., & Gudergan, S. P. (2017). Advanced issues in partial least squares structural equation modeling. Sage publications.

  1. Discussion

This part can be integrated with implications sections.

Action/Answer: Thank you so much for your kind suggestion. We would like to separate the discussion part out of the theoretical and practical implications and limitations, because we would like to discuss in the depth findings and give the reasons to finding.  As well as the clarity of the reader in the study and understanding of each part and preventing confusion in the application of the findings.

7/8. Theoretical implications / Practical implications

This part should be modified to provide more detailed and clear implications.

Action/Answer: Thank you so much for your kind suggestion. Revision has been made on page 15 and 16, line 464 to 491 as below:

Findings revealed that problems with performance appraisal on both the process and on the person have an inverse influence on JS and OC and OCB. This result can be explained by the equity theory. This states that, when people are treated unfairly, their JS reduces and they do not commit to organizations and do not manifest OCB. However, the influence of JS and OC on OCB can also be explained using the exchange theory. If organizations or administrators ensure that their employees are satisfied with their job, then positive OCB will result.

Administrators and human resource managers may use the findings of this study to effectively explain the phenomenon of OCB. Results indicate that unfairness of performance appraisal leads to problems with the process and the appraiser that have negative effects on OCB, JS and OC. Therefore, administrators should design the process of performance appraisal with fairness, transparency, and testability. They should also determine concrete indicators, a proper period of appraisal time, and use an appraisal form suitable for the employees’ duties. In addition, both appraisers and appraisees should be educated through development and training about targets, objectives, and unbiased ratings in order to increase the fairness of performance appraisal.

The results of the study demonstrated that problems in performance appraisals had negative effects directly and indirectly on organizational citizenship behaviors. Therefore, administrators and human resource departments should pay attention to solve these problems by adjusting the system and designing the process to solve all encountered problems. In addition, development of personnel related to performance appraisals should be arranged for these people to know, understand and be capable in testing and rating work performance. This is for performance appraisals to enhance employees’ job satisfaction, organizational engagement, and good citizenship behaviors.

The study results revealed that JS and OC function as mediators concerning problems with performance appraisal and OCB of employees. Therefore, administrators and human resource units should enhance and support employees to love and commit to organizations by creating an atmosphere that facilitates devotion of physical and mental efforts. Moreover, procurement, promotion and support may be improved by using incentives to enhance employees’ JS and effective work operation.

Round 2

Reviewer 2 Report

Thank you for your sincere efforts in the revision process. The revised manuscript has certainly improved from the previous version. I applaud the authors for their hard work. Hope your other works also going well.

Author Response

Dear reviewer:

Thank you for the opportunity to revise our manuscript, Mediating effects of job satisfaction and organizational commitment between problems with performance appraisal and organizational citizenship behavior (JOItmC-883259). We appreciate the careful review and constructive suggestions. It is our belief that the manuscript is substantially improved after making the suggested edits.

Following this letter are the editor comments with I responses in red. The revision has been developed in consultation with my advisor and gurus.

Thank you for your consideration.

This is good.
But let us update last time this paper for others to read this paper a lot

First, Let us change the title as follows

"Job satisfaction, Organizational commitment, and Organizational behavior citizenship: with the implication for motivation of open innovation" You can change the detail of this title. 

Action/Answer: Thank you so much for your kind suggestion. The article’s title cannot be changed to another name because we have received research funding from the Faculty of Business Administration, Rajamangala University of Technology Thanyaburi. Consequently, we are afraid of conflict with the funding agency in future. Could you please let us use the old title?

Second, 
Change the paper structure as follows
1. Introduction
1.1. research question <---1
1.2. reseach object, scope and method <----2

2. Literature reivew and research framewwork
2.1. Literature reviews <-- 3 , major part of 3
2.2. Research framework <-- 3, minor part of 3

3. Analysis 
3.1. Descpriptive analysis <-- 4
3.2. Data analysis <--5

4. Discussion ; the implication for the motivation of open innovation 

5. Conclustion
5.1. the summary of this study <- 6. discussion
5.2 . impcation <-- part of 7
5.3. Limitations and future research topic <---- part of 7 

You can chance the details by yourself

Action/Answer: Thank you so much for your kind suggestions. The manuscript has been altered according to your recommendations.

Third.
Please add "4. Discussion ; the implication for the motivation of open innovation " with following references.
- The open innovation journey: how firms dynamically implement the emerging innovation management paradigm ?
- The culture for open innovaiton dynamics
- Entrepreneurial cyclical dynamics of open innovation
- Micro and macro dynamics of of open innovation with a Quadruple-helix model
- Collective intelligence: an emerging world in open innovation
- Unravelling the process from closed to open innovation: evidence from mature asset-intensive industries

Action/Answer: Thank you so much for your kind suggestions. Revision has been discussed at the end of the Discussion section and changes that you recomend have been added to the article.

In addition, JS and OC will also contribute to the development of organizational open innovation [1],  whereby open innovation will help the organization’s use of inflows and outflows of knowledge to accelerate internal innovation and expand the markets for external use of innovation [2-4]. Therefore, performance appraisal affects JS, OC and OCB, which are considered to be the result of a person's behavior. These factors inevitably affect the emergence of open innovation and lead to the prosperity of the organization [5,6].

  1. Chiaroni, D.; Chiesa, V.; Frattini, F. The Open Innovation Journey: How firms dynamically implement the emerging innovation management paradigm. Technovation 2011, 31, 34-43.
  2. Yun, J.J.; Won, D.; Park, K. Entrepreneurial cyclical dynamics of open innovation. Journal of Evolutionary Economics 2018, 28, 1151-1174.
  3. Chiaroni, D.; Chiesa, V.; Frattini, F. Unravelling the process from Closed to Open Innovation: evidence from mature, asset‐intensive industries. R&d Management 2010, 40, 222-245.
  4. Yun, J.J.; Liu, Z. Micro-and macro-dynamics of open innovation with a quadruple-helix model. 2019, 11, 330, doi:doi:10.3390/su11123301.
  5. Yun, J.J.; Zhao, X.; Jung, K.; Yigitcanlar, T. The Culture for Open Innovation Dynamics. Sustainability 2020, 12, , doi:doi:10.3390/su12125076.
  6. Yun, J.J.; Jeong, E.; Zhao, X.; Hahm, S.D.; Kim, K. Collective intelligence: An emerging world in open innovation. Sustainability 2019, 11, 4495.